

# The relationship between the Functional Movement Screen and the Y Balance Test in youth footballers

Damian Sikora and Pawel Linek

Institute of Physiotherapy and Health Sciences, The Jerzy Kukuczka Academy of Physical Education, Katowice, Poland

## ABSTRACT

**Background:** The Functional Movement Screen (FMS) and the Y Balance Test (Y-BT) are screening tools to detect movement deficits and to identify footballers at high risk of injury. If these tools are able to identify athletes with high risk of injury, they should measure the same construct and also be highly correlated.

**Objectives:** The aim of the study was to determine the relationship between the FMS and Y-BT in youth footballers. The present study also aimed to assess the degree of association between the FMS and Y-BT considering high-injury-risk (FMS <= 14 points and Y-BT <= 89.6%) and low-injury-risk groups (FMS > 14 points and Y-BT > 89.6%).

**Method:** A sample of 226 healthy athletes (mean age: 14.0 ± 2.3 years) was selected from a football club. The FMS and Y-BT data were collected from all participants. The Y-BT raw data were normalised to the relative length of the lower limbs. Spearman's correlation was used in the analysis.

**Results:** For the whole group, there was a moderate correlation (R = 0.41; $p < 0.001$) between the composite FMS score and composite Y-BT score. The strength of relationships varied from weak to moderate between the FMS subtests and most Y-BT results for each direction. In the high-injury-risk group, there was no correlation (R = 0.11; $p = 0.61$) between the composite FMS score and composite Y-BT score. For the low-injury-risk group, there was a weak significant correlation (R = 0.27; $p < 0.007$) between the composite FMS score and composite Y-BT score. Additionally, 56 and 53 athletes were classified to the high-injury-risk group based on the FMS and Y-BT, respectively. Only 23 athletes were classified to the high-risk group by both tests.

**Conclusions:** Youth footballers showed only weak to moderate correlations between the FMS and the Y-BT. Footballers classified in the high-injury-risk group based on the FMS and Y-BT presented a different relationship between the FMS and Y-BT tasks compared to the low-injury-risk group. The results confirmed that the FMS and Y-BT should not be used interchangeably as they assess different movement deficits in the group of youth football players. The study results may partially suggest that using one of these screening tools cannot successfully predict injury risk in adolescent football players. This justifies the need to use these tests simultaneously to identify possible neuromuscular control deficits in youth footballers.

Corresponding author
Pawel Linek,
Linek.fizjoterapia@vp.pl

# INTRODUCTION

The Functional Movement Screen (FMS) and the Y Balance Test (Y-BT) are movement screening tools commonly used in high-level (*Kiesel, Butler & Plisky, 2014*) and youth athletes (*Chalmers et al., 2017*). The FMS has been used to assess neuromuscular control, trunk stability, and side-to-side asymmetry (*Cook, Burton & Hoogenboom, 2006a*; *2006b*; *Bonazza et al., 2017*; *Chalmers et al., 2017*). The Y-BT, on the other hand, has been used to assess dynamic balance and side-to-side asymmetry (*Plisky et al., 2009*). Both tests are also commonly used in scientific research as injury prediction tools for athletes. Some authors have suggested that a composite FMS score below or equal to 14 indicates a high risk of non-contact injury (*Kiesel, Plisky & Voight, 2007*; *Bonazza et al., 2017*). In turn, a normalised composite score below 89.6% in the Y-BT is associated with an increased risk of non-contact injury (*Butler et al., 2013*). If both movement screening tools are really able to identify athletes with high risk of non-contact injury, they should measure the same construct and also be highly correlated.

To date, a weak relationship between the FMS and Y-BT in adult athletes and non-athletes has been shown (*Lockie et al., 2015*; *Kelleher et al., 2017*; *Harshbarger, Anderson & Lam, 2018*). In the literature, however, there are only two studies that have shown a moderate relationship between the FMS and Y-BT in youth athletes (*Kramer et al., 2019*; *Chang et al., 2020*). Thus, there may be a higher correlation between the FMS and Y-BT in adolescents than in adults. *Kramer et al. (2019)* have suggested that both tools measure similar underlying constructs in high school athletes. Similarly, *Chang et al. (2020)* have suggested that youth athletes present similar movement patterns to complete each FMS and Y-BT task. Nevertheless, there is no strong evidence to support such a statement. Additionally, *Kramer et al. (2019)* and *Chang et al. (2020)* have evaluated athletes representing different sports. In fact, *Chang et al. (2020)* have examined the relationship between the FMS and Y-BT in volleyball, basketball, and handball players, whereas *Kramer et al. (2019)* did not specify the group examined in terms of sport practice. To the best of our knowledge, no studies have analysed the relationship between the FMS and Y-BT in youth football players. Some studies have highlighted that each sport exhibits certain movement patterns that are specific to that sport (*Hopkins et al., 2007*; *Whittaker et al., 2017*). It is indicated that injury risk estimation based on the FMS or Y-BT is specific to individual sports (*Butler et al., 2013*; *Chimera, Smith & Warren, 2015*; *Moran et al., 2017*). From this perspective, the relationship estimation between the FMS and Y-BT from other studies should not be applied to youth footballers.

Since the FMS and Y-BT are used as screening tools to detect movement deficit and to identify youth footballers at higher risk of injury (*Butler et al., 2013*; *Chalmers et al., 2017*), it seems important to determine whether there is a relationship between the two tests. More specifically, we were interested to check whether youth football players exhibit similar movement patterns when performing particular FMS and Y-BT movement tasks. From the perspective of using these tests in future studies, it is crucial to be aware of how

complementary these tests can be. Thus, the main aim of this study was to determine the relationship between the individual movement task and composite FMS and Y-BT results in youth footballers. Unlike other studies, the present study also aimed for the first time to assess the strength of association between the FMS and Y-BT considering high- and low-injury-risk groups based on the FMS and Y-BT. The relationship between the FMS and Y-BT may vary according to the presence of compensatory movements or poor performance. In other words, we hypothesised that the relationship between the two tests may be different in the low-injury-risk group compared to the high-injury-risk group.

## MATERIALS AND METHODS

### Study participants and location

The study was conducted at two football academies in the Silesian region of Poland. The study was authorized by the Bioethics Committee for Scientific Studies at the Academy of Physical Education in Katowice (Consent Number 4/2017). All participants and their parents were informed about the procedures performed and provided written informed consent to participate in the study.

Footballers from the Football Academy as well as at the Sport and Recreation Centre in Będzin and Sosnowiec were considered for the study. After reviewing club documents and coaches' information, we invited all male individuals aged 10–17 years who had been regularly attending football training for a minimum of 2 years at least twice a week prior to the start of this study. In total, 240 footballers agreed to participate and were recruited for their health status observation. In order to verify athletes' health status, the 4-month observation on injury and health issues were obtained through the weekly collection of the Oslo Sport Trauma Research Centre (OSTRC) questionnaire (*Clarsen et al., 2014*). Finally, we performed the FMS and then a Y-BT (always in this order) with all participants who met the following basic criteria of eligibility: (a) all players had to be free of any health or injury issues at the time of testing; and (b) having had no injury excluding them for more than a week from training units in the last 4 months prior to the study. Both tests were conducted on the same day for each athlete. All participants were examined during the 12 consecutive days in a closed space (sports hall). During the examinations, the athletes were in a preparation phase for the next football season.

### The functional movement screen (FMS)

The FMS tool consists of seven tests covering basic movement patterns comprising: deep squat; hurdle step; in-line lunge; shoulder mobility assessment; active straight-leg raise; trunk stability push-up; and trunk rotary stability test. Each task performance was evaluated on a 0–3 scale. Each subject performed each task three times as recommended by *Cook, Burton & Hoogenboom (2006a, 2006b)*. A score of 3 means a flawless execution of the trial, a score of 2 is assigned when a compensation of the movement pattern occurred, one point is awarded when the subject was unable to perform a given task correctly and 0 points were assigned when pain occurred at any time during the movement. During asymmetrical tests, the right and left sides were evaluated separately and the lower score obtained during a given task was recorded. The right limb was always evaluated first. After

all the tests were performed, the number of points obtained was summed up, which gave the composite FMS score. The whole FMS procedure, movement tests and their assessment are well explained elsewhere (*Cook, Burton & Hoogenboom, 2006a*, *2006b*).

The results of individual tests were also presented in groups, which included: $FMS_{STABIL.}$ (trunk stability push-up, trunk rotary stability test), $FMS_{FLEX}$ (shoulder mobility assessment, active straight-leg raise) and $FMS_{MOVE}$ (deep squat, hurdle step, in-line lunge) (*Linek et al., 2019*). FMS demonstrates excellent inter-rater and intra-rater reliability (*Bonazza et al., 2017*). Additionally, the intraclass correlation coefficient for intra-rater reliability was 0.81 (95% CI [0.69–0.92]) and 0.81 (95% CI [0.70–0.92]) for inter-rater reliability (*Bonazza et al., 2017*). Following the guidelines of *Bonazza et al. (2017)*, a high (with a composite FMS score <= 14 points) and low-injury-risk (a composite FMS score > 14 points) group were selected.

### Y balance test (Y-BT)

The Y-BT (*Plisky et al., 2009*) was used in the study. The measurement tool consists of a central stand, which is made of plastic, and three tubes are placed in the anterior, posterolateral and posteromedial directions. On each of the tubes there is a movable pointer indicating the measurement accuracy to 0.5 cm. The test begins with the athlete positioned at the centre stand, their upper limbs placed on the hip bone plates. Each subject started the test with their dominant lower limb, always in the same order (direction): anterior, posterolateral, posteromedial. Each athlete performed four training trials, then five measurement trials as recommended by *Linek et al. (2017)*, whose procedure has been developed for youth footballers and showed good reliability ($ICC_{3.1}$ 0.66–0.82-in all directions). The result was not recorded if the subject lost their balance during the trial, pulled a lower limb away from the central stance, pulled their hands away from the hip plates or touched the ground while returning to the starting position. After the test, the relative length of the lower limb (anterior superior iliac crest–medial ankle) was measured on the therapeutic couch using a centimetre tape. The measurement was used to calculate the distance value according to the formula (distance obtained in the test/ relative length of the lower limb) *100 (*Linek et al., 2017*). Following the guidelines of *Butler et al. (2013)*, a high-(with a composite Y-BT score <= 89.6%) and low-injury-risk (a composite Y-BT score > 89.6%) group were selected. Detailed description of the Y-BT procedure is explained in *Linek et al. (2017)* study.

### Body mass and height assessment

A SECA model 799 (SECA, Hamburg, Germany) was used to assess body mass and body height. During measurement, all subjects were barefoot and shirtless.

### Statistical analysis

All analyses were calculated using the Statistica 13.1 package (StatSoft, Tulsa, OK, USA). Basic anthropometric data (mean ± standard deviation) were presented for the entire study group, and for both high-injury-risk and low-injury-risk groups. Injury-risk groups were created by using results from both tests used. Thus, the high-injury-risk group consisted of

**Table 1 Basic data of the population studied.**

| | Football players | | |
| --- | --- | --- | --- |
| | Whole group ($n$ = 226) | High risk[1] group ($n$ = 23) | Low risk[2] group ($n$ = 203) |
| Age (years) | 14.0 ± 2.3 | 13.7 ± 2.7 | 14.0 ± 2.3 |
| Weight (Kg) | 53.6 ± 14.2 | 57.3 ± 14.7 | 53.2 ± 14.1 |
| Height (Cm) | 134.7 ± 63.1 | 136.2 ± 64.6 | 134.5 ± 63.1 |
| BMI (Kg/m$^2$) | 19.5 ± 6.0 | 20.2 ± 2.5 | 19.4 ± 6.3 |
| Years of training | 5.7 ± 1.9 | 5.1 ± 2.2 | 5.8 ± 1.9 |

Notes:
[1] Athletes with a total FMS score <= 14 points and Y-BT <= 89.6%.
[2] Group of athletes with a total FMS score >14 points and Y-BT > 89.6%.
BMI, Body Mass Index; $n$, number of participants.

participants with the composite FMS score <= 14 points and Y-BT <= 89.6%, whereas the low injury-risk-group consisted of participants with the composite FMS score > 14 points and Y-BT > 89.6%. Correlations were assessed between the FMS total score (and the FMS$_{MOVE}$, FMS$_{STABIL}$ and FMS$_{FLEX}$ subscores), and the standardised distance achieved at Y-BT in each direction. Due to the non-normal data distribution in the Shapiro–Wilk test, the researchers decided to use the non-parametric Spearman correlation ($p$-values < 0.05 were considered significant). The Spearman rank correlation coefficient (R) was interpreted according to *Hopkins et al. (2009)*. An R value of 0 to 0.30 or 0 to −0.30 was considered a weak correlation; 0.31 to 0.50 or −0.31 to −0.50 a moderate correlation; 0.51 to 0.70 or −0.51 to −0.70 a strong correlation and 0.71 to 1 or −0.71 to −1 a very strong correlation.

## RESULTS

A retrospective analysis of OSTRC has shown that out of 240 athletes, 226 met the inclusion criteria (full participation in training and competitions, no reduction in training volume, no health complaints). Descriptive statistics of the included participants are shown in Table 1. The data are presented for the whole group and for the high- and low-injury-risk subgroups. Figure 1 illustrates the percentage of participants classified as the high-injury-risk group based on the FMS (56 athletes) and Y-BT (53 athletes). Only 23 participants were classified to the high-injury-risk group in both the FMS and Y-BT. Spearman correlations between the FMS and Y-BT for the whole group, low-risk group, and high-risk group are presented in Tables 2–4, respectively.

For the whole group, there was a moderate significantly positive correlation between the composite FMS score and composite Y-BT score (Fig. 2). Additionally, there were significant positive correlations between the FMS subtests (FMS $_{MOVE}$, FMS $_{STABIL}$, and FMS$_{FLEX}$) and most Y-BT results for each direction. The strength of relationships varies from weak to moderate.

For the low-risk group, there was a weak significantly positive correlation between the composite FMS score and composite Y-BT score (Fig. 3). Additionally, there were weak significant positive correlations between the FMS$_{MOVE}$ and most Y-BT results for each

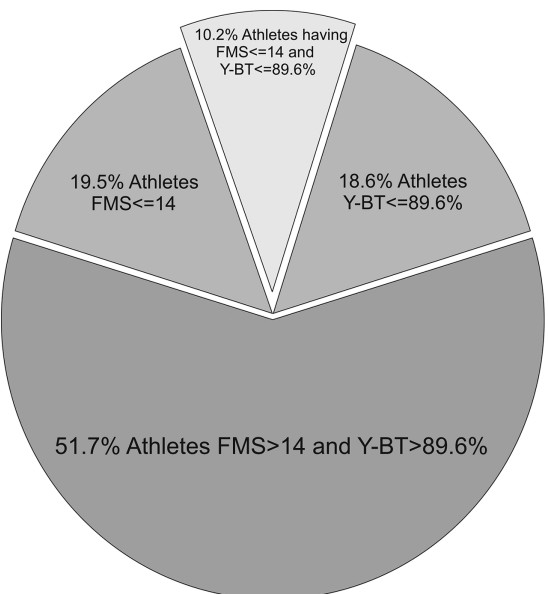

**Figure 1 Percentage of participants classified to high and low injury-risk group based on the FMS and Y-BT.**

**Table 2 Correlation between the Y-BT and FMS in the whole study group.**

|  | FMS total score | p value | FMS move | p value | FMS stabil | p value | FMS flex | p value |
|---|---|---|---|---|---|---|---|---|
| Y-anterior–right | 0.39 | <0.001 | 0.4 | <0.001 | 0.12 | 0.081 | 0.26 | <0.001 |
| Y-anterior–left | 0.42 | <0.001 | 0.41 | <0.001 | 0.18 | 0.007 | 0.27 | <0.001 |
| Y–anterior–average | 0.44 | <0.001 | 0.43 | <0.001 | 0.17 | 0.011 | 0.28 | <0.001 |
| Y–posterolateral–right | 0.31 | <0.001 | 0.27 | <0.001 | 0.18 | 0.007 | 0.21 | 0.001 |
| Y–posterolateral–left | 0.34 | <0.001 | 0.27 | <0.001 | 0.26 | <0.001 | 0.18 | 0.006 |
| Y–posterolateral–average | 0.35 | <0.001 | 0.29 | <0.001 | 0.24 | <0.001 | 0.2 | 0.002 |
| Y–posteromedial-right | 0.27 | <0.001 | 0.26 | <0.001 | 0.12 | 0.073 | 0.17 | 0.009 |
| Y–posteromedial–left | 0.31 | <0.001 | 0.29 | <0.001 | 0.17 | 0.01 | 0.18 | 0.008 |
| Y–posteromedial–average | 0.31 | <0.001 | 0.29 | <0.001 | 0.15 | 0.02 | 0.19 | 0.005 |
| Y–total score | 0.41 | <0.001 | 0.38 | <0.001 | 0.22 | 0.001 | 0.25 | <0.001 |

direction. The $FMS_{STABIL}$ was not related with most Y-BT results, whereas the $FMS_{FLEX}$ was only related with Y-BT results for the anterior direction (Table 3).

For the high-risk group, there was no correlation between the composite FMS score and composite Y-BT score (Fig. 4). The only significant correlations were between the $FMS_{STAB}$ (inverse moderate relationship), the $FMS_{FLEX}$ (positive moderate relationship), and Y-BT result in the anterior direction for the right leg (Table 4).

## DISCUSSION

To the best of our knowledge, this is the first study to analyse the relationship between composite FMS score and composite Y-BT test score, as well as subtests using both

**Table 3 Correlation between the Y-BT and FMS in the low risk group.**

|  | FMS total score | p value | FMS move | p value | FMS stabil | p value | FMS flex | p value |
|---|---|---|---|---|---|---|---|---|
| Y-anterior–right | 0.30 | <0.001 | 0.32 | <0.001 | 0.06 | 0.431 | 0.16 | 0.019 |
| Y-anterior–left | 0.33 | <0.001 | 0.36 | <0.001 | 0.11 | 0.132 | 0.18 | 0.013 |
| Y–anterior–average | 0.35 | <0.001 | 0.36 | <0.001 | 0.1 | 0.159 | 0.18 | 0.01 |
| Y–posterolateral–right | 0.19 | 0.008 | 0.17 | 0.014 | 0.08 | 0.239 | 0.12 | 0.101 |
| Y–posterolateral–left | 0.22 | 0.001 | 0.19 | 0.007 | 0.17 | 0.018 | 0.07 | 0.323 |
| Y–posterolateral–average | 0.22 | 0.002 | 0.19 | 0.005 | 0.14 | 0.048 | 0.08 | 0.23 |
| Y–posteromedial-right | 0.16 | 0.027 | 0.18 | 0.009 | 0.01 | 0.897 | 0.07 | 0.313 |
| Y–posteromedial–left | 0.18 | 0.01 | 0.19 | 0.007 | 0.05 | 0.464 | 0.07 | 0.346 |
| Y–posteromedial–average | 0.17 | 0.013 | 0.2 | 0.005 | 0.03 | 0.656 | 0.07 | 0.315 |
| Y–total score | 0.27 | <0.001 | 0.28 | <0.001 | 0.1 | 0.144 | 0.12 | 0.087 |

**Table 4 Correlation between the Y-BT and FMS in the high risk group.**

|  | FMS total score | p value | FMS move | p value | FMS stabil | p value | FMS flex | p value |
|---|---|---|---|---|---|---|---|---|
| Y-anterior–right | 0.21 | 0.336 | 0.29 | 0.179 | −0.51 | 0.013 | 0.46 | 0.026 |
| Y-anterior–left | −0.24 | 0.28 | −0.21 | 0.333 | −0.26 | 0.233 | 0.25 | 0.256 |
| Y–anterior–average | 0.01 | 0.987 | 0.04 | 0.846 | −0.41 | 0.051 | 0.39 | 0.065 |
| Y–posterolateral–right | 0.24 | 0.262 | 0.08 | 0.707 | 0.12 | 0.571 | 0.03 | 0.904 |
| Y–posterolateral–left | −0.05 | 0.833 | −0.3 | 0.168 | 0.12 | 0.584 | 0.08 | 0.729 |
| Y–posterolateral–average | 0.07 | 0.739 | −0.12 | 0.588 | 0.09 | 0.691 | 0.07 | 0.761 |
| Y–posteromedial-right | 0.11 | 0.615 | −0.13 | 0.547 | 0.36 | 0.096 | −0.14 | 0.515 |
| Y–posteromedial–left | −0.08 | 0.729 | −0.19 | 0.374 | 0.24 | 0.274 | −0.18 | 0.423 |
| Y–posteromedial–average | 0.03 | 0.875 | −0.21 | 0.326 | 0.34 | 0.111 | −0.15 | 0.494 |
| Y–total score | 0.11 | 0.612 | −0.13 | 0.563 | −0.01 | 0.974 | 0.23 | 0.285 |

screening tools in youth footballers. It was also decided for the first time to assess the relationship between both tests in low- and high-injury-risk groups. The results for all participants showed a weak to moderate association between both tests. The FMS subtests ($FMS_{MOVE}$, $FMS_{STABIL}$, and $FMS_{FLEX}$) were also related to results from most Y-BT reach distances for each direction. In the low-injury-risk group, there were similar results as for the whole group, but the $FMS_{STABIL}$ was not related to the Y-BT at all. In the high-injury-risk group, in turn, there was no correlation between the composite FMS score and composite Y-BT score, and moderate correlations between some FMS subtests ($FMS_{STAB}$ and $FMS_{FLEX}$) and anterior distance obtained in the Y-BT. However, in the high-risk group, there inverse relationships were also detected, suggesting that athletes with a higher FMS score had a lower score in the Y-BT. This was unexpected but partially confirmed our initial hypothesis that the relationship between the two tests is different in the low-injury-risk group compared to the high-injury-risk group. Such results mean that in both

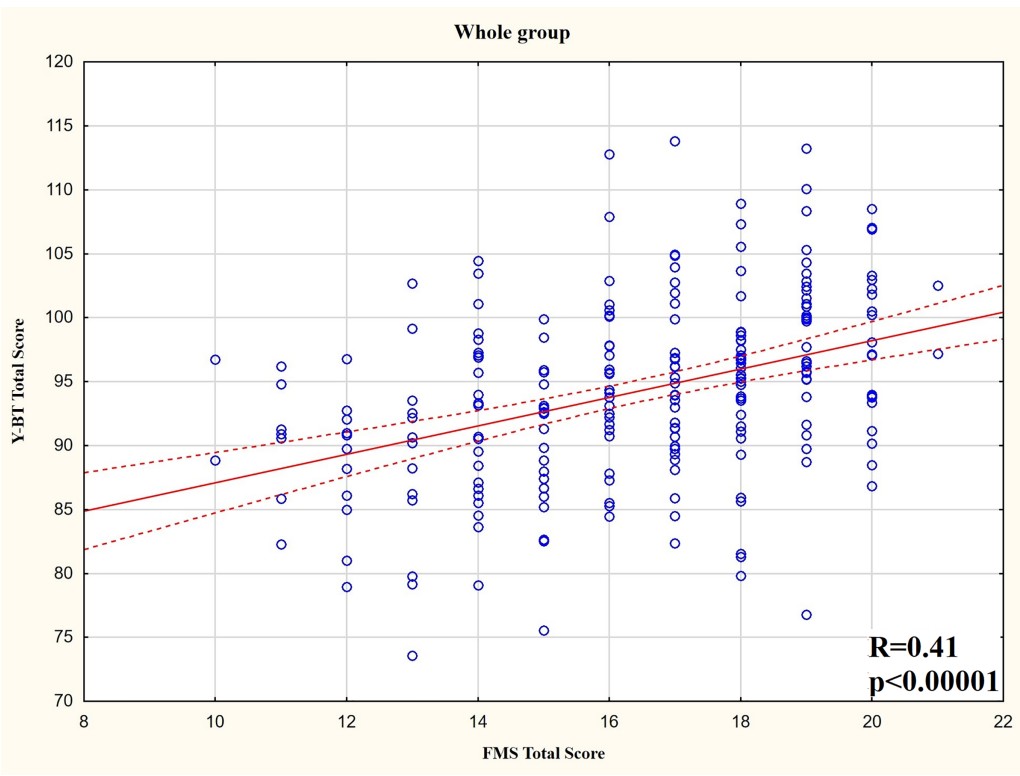

**Figure 2 Y-BT composite score and FMS composite score distribution for the whole group.**

screening tools there were some movement tasks related to each other in different ways in athletes with poor movement performance or patterns seen in both tests. Additionally, the strength and direction of the relationship between both tests suggested that the FMS and Y-BT measure different constructs. This was also confirmed by results showing that only 41.1% of adolescent athletes presenting high risk of injury in the FMS were also detected as high risk in the Y-BT. Such observation has suggested that at least one of these screening tools cannot successfully predict injury risk in adolescent football players. Therefore, although the FMS and the Y-BT were designed to assess movement deficits and asymmetry, they should not be used interchangeably.

As the assessment of the relationship between the FMS and Y-BT in previous studies (*Lockie et al., 2015*; *Kramer et al., 2019*; *Chang et al., 2020*) did not take into account the injury-risk group, it is difficult to conclude whether the present study results are specific to the participants examined (football players) or can be applied to the general population. Our results clearly confirmed the suggestion of *Lisman et al. (2021)* that the FMS and Y-BT should not be used as stand-alone tools to assess injury risk in youth athletes. The strength of correlation between the FMS and Y-BT obtained in the present study suggested that athletes with high (or low) FMS scores did not necessarily have proportionally high (or low) Y-BT scores. The weak correlation between the two tests is also indicated by the low percentage of adolescent athletes presenting high risk of injury in the FMS and Y-BT simultaneously. Thus, there is a reason to claim that the FMS and Y-BT evaluate slightly
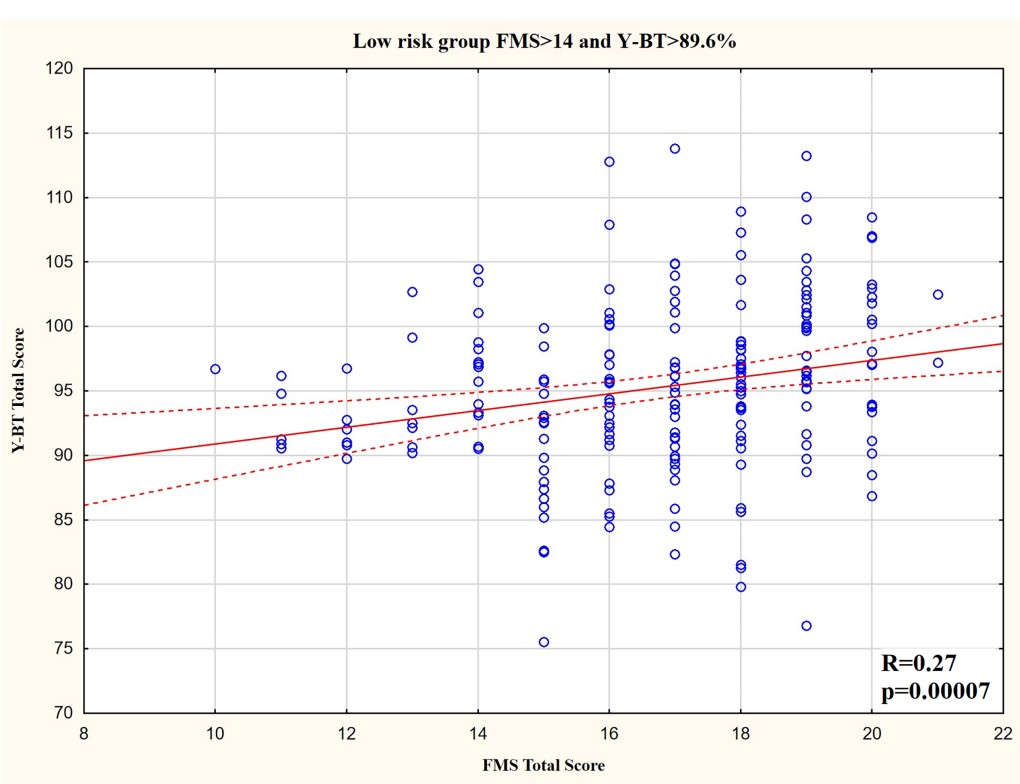

**Figure 3 Y-BT composite score and FMS composite score distribution for the low injury risk group.**

different (unrelated) motor deficits in youth football players. From this perspective, they may be more complementary. It is also worth noting that analysis of correlation in the low- and high-injury-risk groups taking into account FMS subtests ($FMS_{MOVE}$, $FMS_{FLEX}$, $FMS_{STABIL}$) allowed unexpected results to be seen, like an inverse relationship between the $FMS_{STAB}$ and Y-BT result in the anterior direction. Some researchers have suggested that an analysis of individual FMS tasks (FMS subtests–$FMS_{MOVE}$, $FMS_{FLEX}$, $FMS_{STABIL}$) helps to better understand the movement deficits in footballers (*Marques et al., 2017*). The present study results confirmed this suggestion.

Taking into account the whole group, the strength of association between the FMS and Y-BT (or the Star Excursion Balance Test (SEBT)) in the present article was similar to other studies (*Lockie et al., 2015*; *Harshbarger, Anderson & Lam, 2018*). *Lockie et al. (2015)* obtained a moderate relationship between individual FMS tasks and SEBT results in adult athletes. *Harshbarger, Anderson & Lam (2018)* obtained only a moderate correlation between the trunk rotary stability test and the anterior direction of the SEBT test (and a weak correlation with the posteromedial direction) in adult athletes. Although, in the aforementioned studies, an older version of the Y-BT was used (the SEBT), and only adults (female and male) were analysed, similar results were obtained. However, we obtained different results than those presented by *Chang et al. (2020)*, where higher relationship strength was obtained between individual FMS tests and Y-BT in a group of youth athletes. *Chang et al. (2020)* examined volleyball, basketball and handball players, while the present

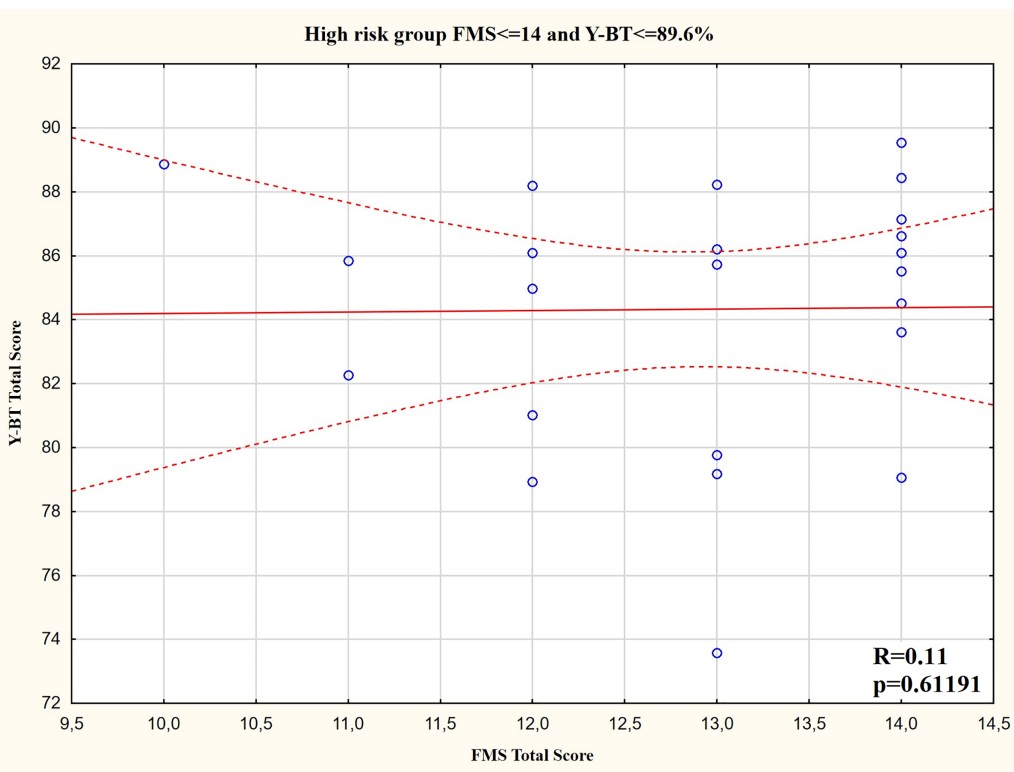

**Figure 4 Y-BT composite score and FMS composite score distribution for the high injury risk group.**

study examined a homogeneous group of football players. Each sport is characterised by movement patterns typical only for that sport (*Hopkins et al., 2007*; *Whittaker et al., 2017*). From this perspective, there is a possibility that there are sports in which the strength of correlation between the FMS and Y-BT is higher, and for such sports it would be justified to claim that these two tests measure similar constructs. However, additional similar studies need be conducted, including separate analysis of each sports discipline, because otherwise such a hypothesis cannot be verified.

It is also worth mentioning the limitations of the present study. The age of puberty was not assessed in the study group, and the wide age range of 10–17 years suggests that the participants were at different stages of puberty (pre-puberty, post-puberty, and during puberty) and have different experience in sport practice. Only adolescent football players were included in the study, so the results obtained should not be freely transferable to a group of women or children, or to youths involved in other sports or non-athletes.

## CONCLUSIONS

Youth footballers showed only weak to moderate correlations between the FMS and the Y-BT. Footballers classified in the high-injury-risk group based on the FMS and Y-BT presented a different relationship between the FMS and Y-BT tasks compared to the low-injury-risk group. The obtained results confirmed that the FMS and Y-BT should not be used interchangeably as they assess different movement deficits in the group of youth

football players. The study results may also partially suggest that using only one of these screening tools cannot successfully predict injury risk in adolescent football players. This justifies the need to use these tests simultaneously to identify possible neuromuscular control deficits in youth footballers.

### Funding

The study was funded by the Team of Biomedical Basis of Physiotherapy, The Jerzy Kukuczka Academy of Physical Education in Katowice The funders had no role in study design, data collection and analysis, decision to publish, or preparation of the manuscript.

### Grant Disclosures

The following grant information was disclosed by the authors:
Team of Biomedical Basis of Physiotherapy.
The Jerzy Kukuczka Academy of Physical Education in Katowice.

### Competing Interests

The authors declare that they have no competing interests.

### Author Contributions

- Damian Sikora conceived and designed the experiments, performed the experiments, analyzed the data, prepared figures and/or tables, authored or reviewed drafts of the article, and approved the final draft.
- Pawel Linek conceived and designed the experiments, analyzed the data, authored or reviewed drafts of the article, and approved the final draft.

### Human Ethics

The following information was supplied relating to ethical approvals (*i.e.*, approving body and any reference numbers):

The study was approved by the Bioethics Committee for Scientific Studies at the Academy of Physical Education in Katowice (4/2017).

### Data Availability

The raw measurements are available as a Supplemental File.

### Supplemental Information

Supplemental information for this article can be found online at http://dx.doi.org/10.7717/peerj.13906#supplemental-information.

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
