# Peer review of "The relationship between the Functional Movement Screen and the Y Balance Test in youth footballers"

_PeerJ, doi:10.7717/peerj.13906_

## Round 0.1 · original submission · Major Revisions

Dear Authors,

The manuscript needs major revisions to improve.
Please reply point by point to the reviewers' comments.

Reviewer 1 ·

Basic reporting

The article has major issues with language and clarity in presenting their rationale and results. It will need a solid re-write before publication.

Experimental design

No comment.

Validity of the findings

No comment.

Additional comments

Comments to Authors
I congratulate the authors on an interesting study! The study main aim is to investigate the relationship between the FMS and Y-BT test.
Although functional movement tests are significantly studied for years now, I think this study has merit in adding new information to the already body of FMS and Y-BT literature. However, the paper will need major revision before publication. There are major issues with language and grammar, and methodological issues.

Main comments

Abstract
1. The abstract is unclear and scarcely written. I’d like the authors to edit the following:
a) The background – please be more precise on the background and aim for the study. Why are these two tests frequently used as a screening tool? Why is it important to investigate whether there’s a relationship between these two tests?
F.ex: L32 screening tools for what? Injuries? To enhance performance?
L33 screening tools for functional deficit. What functional deficit?

2. Objectives – L37 and L38 please specify if the association between FMS high or low risk and Y-BT was with the total score of Y-BT or high or low risk of Y-BT.

3. Method – please specify your participants – male and/or females, one football club and team?
a) L41: language – “The FMS test and Y-BT scores were collected”. This sentence is unclear, please re-write. Do you mean that the players were tested on the FMS and Y-BT at one time-point? Or on separate days?

4. Results and conclusion: individual FMS. Please be consistent that this is the grouped individual test. Please re-phrase.


Introduction

5. The introduction is unclear and poorly written. It is for me unclear what the rationale for the study is. Why is it important to investigate the relationship between these two tests? What is your fact 1 and Fact 2 leading to your aim? I’d like the authors to elaborate more on the characteristics of these two tests – what are the (believed) similarities between FMS and YBT (are they believed to measure the same construct)? What are the differences? What does a good score on FMS mean? And what does a good score on the Y-BT mean? Why did the authors choose to investigate relationship between FMS high and low risk group and Y-BT? I presume Y-BT total score and not Y-BT high and low risk group? Please elaborate and re-write.

Method

6. The description of the participants and inclusion/exclusion criteria are unclear. Please add a detail description of the following:
a) Please state if the participants included males and females
b) Please clarify if also the participants (youth athletes) gave a written informed consent
c) The OSTRC questionnaire on health problems were used – what was the purpose of using this questionnaire as an inclusion criteria? Was it to make sure that the players were free of any health or injury issues at the time of testing? If so, I suggest the authors re-phrase their inclusion criteria to: All players had to be free of any health or injury issues at the time of testing. The information on injury and health issues were obtained by using the OSTRC health questionnaire (Clarsen et al,…). Please explain and add to the method.
d) What was the reason for the second exclusion criteria in L128-129? How was it obtained? Self-reported? Self-reported on a questionnaire? Please explain and add to the method.
e) Please add information on the testing procedure. Was both test conducted on the same day for each athlete? All 226 athletes tested on the same day?

Results

7. The authors have omitted to present demographic results of the population. These results are essential and should not be omitted. Please add these results. F.ex How many athletes were invited to participate? How many had to be excluded due not filling the inclusion criteria? Also, I prefer a presentation of the group as in Table 1 in the result section. Please see my comment above.

8. The result section is generally unclear and poorly written. The result section is lengthy and imprecise. The correlation coefficient R should be presented in the text as the authors have done in their abstract. Please add and I suggest the authors have a look at similar studies and re-write the result chapter all together.

Discussion

9. The authors attempt to discuss their results on grounds of existing literature. I strive to grasp what the authors mean are their main findings, importance of their findings and how they will affect future testing procedure. They do touch on it, but should be more precise and elaborate. F.ex L222-L224 This paragraph is unclear to me. What the authors mean are their main findings? Why is it important to find the relationship between the high and low risk FMS score to Y-BT?

10. The authors use individual FMS to describe their grouped individual FMS test. This is not the same as looking at the 7 individual test. Please re-phrase and be consistent.

11. L275 to L291 Las paragraph – the whole paragraph is very unclear to me. What is the importance of these findings with the low and high injury risk groups and Y-BT? The authors need to elaborate and explain what they include in functional deficits. What will the Y-BT test result give which the FMS (total score and grouped individual tests) would not?

12. L269 “The wide age range of youth athlete in the present study.” However, according to your mean and SD (14±2), your age group is not that different to Wen-Dien Chang et al. Please explain and change accordingly.

General comments

13. F.ex: L61 “The FMS and Y-BT are screening tools commonly used….”. What are FMS and YBT screening tools for? Please elaborate and see my comment in 1a. The authors introduce background on evidence on the relationship of these two tests for then in the third paragraph introduce literature on these tests as an injury risk tool. I’d suggest the authors to move the last paragraph to

14. L84. The authors conclude that “there is no relationship between FMS and YBT, but in youth athletes such relationship may exist”. How can the authors conclude this based on the literature provided in the paragraph above for this sentence (L73 to L83). I disagree, a weak relationship between the two tests in an adult and non-athlete populations cannot be concluded that there’s no relationship. Also, Kramer et al (2019 in Phys Int J Sports Therapy) did find a relationship between the FMS composite score and YBT. Please clarify, re-phrase and reference Kramer et al.

15. L94 “To the best of our knowledge, no studies have analysed the relationship between FMS and Y-BT in football players…” As authors, you should know this. Please re-phrase. F.ex. There is no or limited studies that have…..

16. L96-97 “Both FMS and Y-BT screening tools used as….” Repetitive. Please re-phrase. In general, there are some loose hanging sentences in the introduction. F.ex: L93 Football is one of the main cause of lower limb injuries among youth athletes”. This sentence doesn’t follow on from the previous sentences nor does it lead to the next sentences. Please re-phrase.

17. L109 relationship between FMS and Y-BT total/composite score? Please specify

18. L111 relationship between FMS high and low risk group and Y-BT total score or Y-BT high and low risk group? Please specify. Why did the authors choose to investigate this? Please see my comment #5

19. L112-115 “It may be that………magnitude of the motor deficits of the subjects tested in the FMS test.” This hypothesis is unclear to me. Please explain? And how was this tested?

Method

20. L125 The authors refers to Table 1 as a basic description of the athletes. Table 1 is presenting results and results should never be presented in the method. Please delete the referral to Table 1 in the method and move it to the results. It does not add any value to the text in L125, but will in the result section.

21. L141 Was the L or R leg/arm tested first?

22. L141 Please edit the following: “………,the right and left sides were evaluated separately and the worse score…” to “……lower score obtained….”

23. L147 Please add reference on the reliability of the FMS.

Discussion

24. L228-L230: “..,thus confirming our initial hypothesis that the magnitude of functional deficits obtained in the FMS influences the magnitude of the association of this test with the Y-BT.” This sentence is unclear. What do the authors mean by magnitude of functional deficits? Please re-phrase and explain.

25. L232-L234. Again, unclear sentence. Please re-phrase and explain. What do the authors mean by different movement patterns? What implication has that for injury? Test result? Use of the tests?

26. L240-L241. Language. “It should be assumed that……” Again, unclear sentence. Please re-phrase.

27. L247 SEBT – please write the full name of the test. Abbreviations should never be used unless you have written the full name elsewhere.

Annotated reviews are not available for download in order to protect the identity of reviewers who chose to remain anonymous.

Reviewer 2 ·

Basic reporting

Dear Authors, although the main topic of your study should be of interest, i think you may improve your manuscript before submitting it again. The introduction may be clearer and smoother. You should write 3-5 brief paragraphs, from general to specific information; you should insert your questions and purposes at the end of this section.
The materials and methods section is well written but needs more information on study design and warm-up before motor tests.
I think the results request great improvements. In lines 104-106 you said "A recent systematic review by Pilsky et al. [2021] suggested that both the age and sport type of the examined athletes should be taken into account when identifying an injury factor using the Y-BT", but your analysis just report information by risk groups. In addition, i think you should perform some regression models to make your work more informative. The tables report too many numbers; they could be clearer if you round to at most three decimal places. Finally, you should use the active tense for this section and the text does not report the same information as the tables.

Experimental design

I think you should enforce your analysis to confirm what you said in the discussions. The correlation analysis alone could not give you a cause-effect relationship. Also, more information on several age groups is needed. The maturity status of teenagers constantly mutates and it could affect your results. Finally, in lines 87-89 you wrote "However, the question is whether the degree of relationship between these tests applies to the general population of youth athletes or whether it is sport-specific", but you just report results on football players. Introduction, results, and discussions should share the same topics and follow an accurate order.

Validity of the findings

As previously reported, I think your study is very interesting but you should enforce your analysis and improve your manuscript.

---

## Round 0.2 · Major Revisions

The manuscript needs major revisions to improve.

Please reply point by point to the reviewers' comments.

Reviewer 1 ·

Basic reporting

The manuscript are signficantly improved, but would still benefit from language improvement and word smithing.

Experimental design

no comment

Validity of the findings

no comment

Additional comments

The authors have done a very good job in improving the manuscript based on received comments from the reviewers. I am satisfied with response to my comments and suggest publication without further review.

Reviewer 2 ·

Basic reporting

Dear authors,
although the main focus of your study is very attractive, many evaluations may be done. In my opinion, your analysis is not enough to draw any conclusions about these motor tests in football. As you reported, previous indications said injury risk estimation based on the FMS or Y-BT is specific to individual sports. Thus, it could be interesting to report any regression model using data on injuries and health issues collected with the OSTRC, and the motor test results, in a team sport.

Experimental design

Generally, methods and materials are well described. Ethical standards were applied correctly. The FMS and Y-balance test procedures are clear and accurate. In my opinion, the enrollment process should be clearer and the authors should explain how participants were selected before and recruited. The results show different tables from low to high risk injured participants, but it is not clearly explained how this classification was done. Height and Weight are reported in table 1, but methods lack information on these measurements. In addition, the environmental criteria applied during tests should be reported. Finally, the statistical analysis description is simple and clear.

Validity of the findings

In my opinion, the contents reported are too poor. The sample size is good, but a priori analysis to calculate it was not assessed. The correlations between two or more variables exhibit how one of these varies when one other varies too, but it cannot show any cause-effect relationship. Thus, many confounding external factors could have affected this relationship. Also, it cannot be concluded whether these screening tools could predict (or not) injuries in football; to make it, a different study design and investigation should be assessed.

Additional comments

The type I error probabilities should be of three decimal places. Figure 1 should be omitted, and its results may be reported in the text.

Reviewer 3 ·

Basic reporting

no comment

Experimental design

I will provide suggestions for improving the manuscript step by step

Abstract

Please include further detail about statistical differences and descriptive data.

Introduction

Line 73. Describe the two studies in favor better and more broadly. It would reinforce the study's purpose (Kramer et al. 2019; Wen-Dien Chang et al. 2020).
Line 79. Describe the studies on youth athletes better and more broadly
Methods
Please add further explanation about the moments of data collection. It is not clear how the data was collected, the sequence of the exercises and the time of each collection. Such time could compromise the results.
Line 121. I would suggest to the authors to better describe the evaluation tools (FMS and Y-BT), also using some images/tables of the protocol. It would give more clarity to the whole manuscript. Draw the attention of your work with a figure would be better.

Validity of the findings

Discussion

Line 205. However, in the high-risk group, some of the significant relationships were inversed, suggesting that athletes with a higher FMS score had a lower score in the Y-BT. This is largely speculative
Line 218. As the assessment of the relationship between the FMS and Y-BT in previous studies ……The studies referred to should always be mentioned
I would suggest to the authors to add as a limit of the study also the different experiences of the sample, in sports practice. 10-17 years is a very significant age range

---

## Round 0.3 · accepted · Accept

Congratulations!
The manuscript is ready for publication!

Reviewer 2 ·

Basic reporting

Dear authors,
Your manuscript is clear and concise. A sufficient background is provided and the hypothesis is clear.

Experimental design

Although no recommendations are requested to improve your manuscript, further analysis should be assessed and provided.

Validity of the findings

Conclusions are well stated.

Reviewer 3 ·

Basic reporting

The manuscript is clear and to point.

Experimental design

The authors did a good job of synthesizing the literature.
The aim is clear and to point.

Validity of the findings

The manuscript is clear and to point.